# Changes in Light Energy Utilization in Photosystem II and Reactive Oxygen Species Generation in Potato Leaves by the Pinworm *Tuta absoluta*

**DOI:** 10.3390/molecules26102984

**Published:** 2021-05-18

**Authors:** Ilektra Sperdouli, Stefanos Andreadis, Julietta Moustaka, Emmanuel Panteris, Aphrodite Tsaballa, Michael Moustakas

**Affiliations:** 1Institute of Plant Breeding and Genetic Resources, Hellenic Agricultural Organisation–Demeter (ELGO-Dimitra), 57001 Thermi, Thessaloniki, Greece; stefandr@ipgrb.gr (S.A.); tsaballa80@gmail.com (A.T.); 2Department of Botany, Aristotle University of Thessaloniki, 54124 Thessaloniki, Greece; moustaka@plen.ku.dk (J.M.); epanter@bio.auth.gr (E.P.)

**Keywords:** biotic stress, chlorophyll fluorescence imaging, herbivore insects, hydrogen peroxide, light energy use, non-photochemical quenching, photosynthetic efficiency, PSII photochemistry, singlet oxygen, *Solanum tuberosum*

## Abstract

We evaluated photosystem II (PSII) functionality in potato plants (*Solanum tuberosum* L.) before and after a 15 min feeding by the leaf miner *Tuta absoluta* using chlorophyll *a* fluorescence imaging analysis combined with reactive oxygen species (ROS) detection. Fifteen minutes after feeding, we observed at the feeding zone and at the whole leaf a decrease in the effective quantum yield of photosystem II (PSII) photochemistry (Φ*_PSII_*). While at the feeding zone the quantum yield of regulated non-photochemical energy loss in PSII (Φ*_NPQ_*) did not change, at the whole leaf level there was a significant increase. As a result, at the feeding zone a significant increase in the quantum yield of non-regulated energy loss in PSII (Φ*_NO_*) occurred, but there was no change at the whole leaf level compared to that before feeding, indicating no change in singlet oxygen (^1^O_2_) formation. The decreased Φ*_PSII_* after feeding was due to a decreased fraction of open reaction centers (q*_p_*), since the efficiency of open PSII reaction centers to utilize the light energy (F*v*′/F*m*′) did not differ before and after feeding. The decreased fraction of open reaction centers resulted in increased excess excitation energy (EXC) at the feeding zone and at the whole leaf level, while hydrogen peroxide (H_2_O_2_) production was detected only at the feeding zone. Although the whole leaf PSII efficiency decreased compared to that before feeding, the maximum efficiency of PSII photochemistry (F*v*/F*m*), and the efficiency of the water-splitting complex on the donor side of PSII (F*v*/F*o*), did not differ to that before feeding, thus they cannot be considered as sensitive parameters to monitor biotic stress effects. Chlorophyll fluorescence imaging analysis proved to be a good indicator to monitor even short-term impacts of insect herbivory on photosynthetic function, and among the studied parameters, the reduction status of the plastoquinone pool (q*_p_*) was the most sensitive and suitable indicator to probe photosynthetic function under biotic stress.

## 1. Introduction

Potato (*Solanum tuberosum* L.), introduced to Europe from the Americas during the second half of the 16th century, is extensively cultivated throughout the world, being the world’s fourth largest food crop, after maize, wheat, and rice [1], with a total global cultivation land area of about 20 million hectares [2,3]. The leaf miner *Tuta absoluta* (Meyrick) (Lepidoptera: Gelechiidae) is one of the most harmful phytophagous pests and has restricted the production of tomato worldwide, causing severe problems by reducing yield in both open fields and greenhouse conditions [4,5,6]. Although tomato is regarded as the main host of *T. absoluta*, the pest can also feed, develop, and reproduce on other cultivated Solanaceae, such as *S. tuberosum* [7]. *Tuta absoluta,* first reported in 2006 in Spain and subsequently spread quickly throughout the Mediterranean Basin [5,8], is considered an economically important pest because of its short developmental time, its high reproduction potential, an inadequate management knowledge, and a lack of biological enemies [6].

Evaluation of primary productivity loss by herbivory insects, estimated by the amount of leaf tissue removed, fluctuates from 5% to 30% of yield [9]. Nevertheless, this method does not take into account how herbivory affects photosynthesis of the leftover leaf parts [10]. Photosynthesis in the remaining leaves of the plants can be up-regulated as a mechanism of tolerance to herbivory [11,12] or in most cases can be decreased [13,14,15,16,17,18,19]. Consequently, there appears to be an unpredictability in the effects of herbivory on photosynthesis, which is based on herbivore feeding and plant species.

In comparison to the effects of insect feeding on plant metabolism, little is known about how plant photochemistry responds to herbivore insects. Since the photochemical reactions of photosynthesis provides the energy supply needed for the synthesis of compounds used in defense, such as hormones and primary and secondary defense related metabolites, photosystem II (PSII) photochemistry has to be integrated into the plant’s reaction to herbivory [20]. The absorbed light energy (as photons) by the light-harvesting complexes (LHCs) is transferred to the reaction centers (RCs) where, through charge separation, the electrons flow from photosystem II (PSII) to photosystem I (PSI), resulting on the one hand in the generation of a proton gradient (ΔpH) that drives ATP synthesis and on the other hand in the reduction of NADP^+^ by the electrons transferred [21,22,23,24]. However, under most biotic or abiotic stresses the absorbed light energy exceeds the level that can be used, resulting in an increased formation of reactive oxygen species (ROS), such as superoxide anion radical (O_2_**^•^**^−^), hydrogen peroxide (H_2_O_2_), and singlet oxygen (^1^O_2_) [25,26,27,28,29,30,31,32,33]. Singlet oxygen (^1^O_2_) is formed when unquenched singlet excited states of chlorophyll (^1^Chl*) undergo intersystem crossing and the resulting triplets (^3^Chl*) react with oxygen (O_2_) [34,35,36,37,38]. ROS–antioxidant interactions provide essential information for the redox state that influences gene expression associated with biotic and abiotic stress responses, modulating the appropriate initiation of photosynthetic acclimation or cell death schedules to maximize defense [39,40,41,42,43].

The photochemical reactions of photosynthesis can be assessed by chlorophyll fluorescence analysis in vivo that is extensively used to explore the function of the photosynthetic apparatus and for the assessment of photosynthetic tolerance mechanisms to biotic and abiotic stresses [9,12,44,45]. However, photosynthetic function is not homogeneous at the leaf surface; as a result, standard chlorophyll fluorescence analysis is not characteristic of the photosynthetic status of the whole leaf [45,46,47,48,49,50]. The development of the method of chlorophyll fluorescence imaging overcomes this problem by being capable of identifying spatial heterogeneity of leaf photosynthetic performance at the whole leaf surface and by monitoring early changes in a plant’s physiological status upon early biotic stress cases, before visual symptoms appear [51,52,53]. The chlorophyll fluorescence imaging method is appropriate for visualizing the heterogeneity in plant responses to biotic stresses at an early stage against a background of unaffected plant tissue [54,55].

Besides tomato, *Tuta absoluta* can grow successfully on other alternate hosts such as potato, a close relative of tomato, for which the impact of the pest on the photosynthetic function has not yet been studied extensively [7,56]. In the present work, we examined the impact of the leaf miner, *Tuta absoluta,* on potato photosynthetic function by using chlorophyll fluorescence imaging analysis to study the light energy utilization and photochemical efficiency of photosystem II (PSII), combined with ROS detection, after short feeding duration. We evaluated how herbivory affects the photochemical efficiency of the remaining leaf parts to find out whether a decreased photosynthetic function or an up-regulated photochemical efficiency exists and attempted to gain an insight into the mechanisms that play a role in plant responses to herbivore insects.

## 2. Results

### 2.1. Light Energy Utilization in Photosystem II of Potato before and after Feeding

The changes in light energy utilization in PSII before and after feeding by the leaf miner, *Tuta absoluta,* were estimated by measuring the three parameters of chlorophyll fluorescence, Φ*_PSII_*, Φ*_NPQ_* and Φ*_NO_*, where Φ*_PSII_* represents the effective quantum yield of PSII photochemistry, Φ*_NPQ_* the quantum yield of regulated non photochemical energy loss in PSII, and Φ*_NO_* the quantum yield of non-regulated energy loss in PSII, the sum of all three to be equal to 1 [57].

After 15 min of feeding, an increased spatial heterogeneity was observed (Figure 1). At the feeding zone and at the whole leaf level, Φ***_PSII_*** decreased significantly (Figure 1 and Figure 2a). At the same time, at the feeding zone, Φ*_NPQ_* increased slightly (n.s.) (Figure 1 and Figure 2a), resulting in a significant increase of Φ***_NO_*** at the feeding site (Figure 2a). However, at the whole leaf level, while there was a significant increase of Φ***_NPQ_***, there was no significant change of Φ***_NO_*** (Figure 2a).

### 2.2. Electron Transport Rate, Heat Dissipation, and Open Reaction Centers in Photosystem II of Potato before and after Feeding

The fraction of open reaction centers (q*_p_*) decreased significantly after feeding, at both the feeding zone and the whole leaf level (Figure 1 and Figure 2b). Non-photochemical quenching (NPQ) at the feeding zone did not differ compared to the control, after 15 min of feeding, but was increased significantly at the whole leaf level (Figure 3a). The electron transport rate (ETR) followed the pattern of the effective quantum yield (Φ*_PSII_*), that is, ETR decreased significantly at the feeding zone and at the whole leaf level (Figure 3b).

### 2.3. Efficiency of Photosystem II Photochemistry before and after Feeding

The maximum efficiency of PSII photochemistry (F*v*/F*m*) (Figure 4a) and the efficiency of the water-splitting complex on the donor side of PSII (F*v*/F*o*) (Figure 4b), after 15 min feeding, decreased at the feeding zone but at the whole leaf level did not differ compared to that of control.

Excess excitation energy (EXC) increased significantly after 15 min feeding at the whole leaf level compared to that of control, while at the feeding zone, it was significantly higher compared even to that of the whole leaf level (Figure 5a). The efficiency of open PSII reaction centers (F*v*′*/*F*m*′) was not affected after feeding at both the feeding zone and the whole leaf level (Figure 5b).

### 2.4. Hydrogen Peroxide Detection before and after Feeding

Before feeding, hydrogen peroxide (H_2_O_2_) was slightly detectable only in leaf hairs and in leaf veins, being more visible at the main vein (Figure 6a). After feeding, an increased H_2_O_2_ production was detected, restricted though at the feeding zone area (Figure 6b).

## 3. Discussion

By applying chlorophyll fluorescence imaging analysis, we are beginning now to recognize how photosynthesis is modulated in the undamaged remaining leaf tissue following herbivory [9]. Herbivory is an important selective pressure in most plant species, as it usually results in growth reduction and decreased plant fitness [12]. Our results show that photosynthetic function of potato leaves exhibits clearly differential responses among the feeding zone and at the surrounding areas in response to insect herbivory. After *T. absoluta* feeding, the whole potato leaflets exhibited no statistical differences in the maximum efficiency of PSII photochemistry (F*v*/F*m*) compared to that before feeding, whereas the feeding zone exhibited a significantly lower F*v*/F*m* (Figure 4a). The efficiency of the water-splitting complex on the donor side of PSII (F*v*/F*o*) exhibited the same pattern with F*v*/F*m* (Figure 4b). Both the F*v*/F*m* ratio [58] and its interrelated one F*v*/F*o* [59,60,61] offer an evaluation of the potential PSII efficiency of dark-adapted leaves [55,62]. In our experiment, the decreased F*v*/F*o* ratio at the feeding zone reveals a lower efficiency of the water-splitting complex on the donor side of PSII [63,64], suggesting a donor side photoinhibition mechanism by malfunction of the water-splitting complex [65,66,67,68] that may result to harmful oxidations in PSII [55,68].

The light energy being used in photochemistry (Φ*_PSII_*) at the feeding zone was 23% lower compared to that before feeding, while at the whole leaflet after feeding it was 9% lower (Figure 2a). Since this decrease in Φ*_PSII_* at the feeding zone was not compensated by increases in the photoprotective energy dissipation (Φ*_NPQ_*), it resulted in a high increase in Φ*_NO_* at the feeding zone (Figure 2a). Φ*_NO_* comprises chlorophyll fluorescence internal conversions and intersystem crossings that result in ^1^O_2_ formation via the triplet state of chlorophyll (^3^chl*) [34,35,36,69]. Singlet oxygen (^1^O_2_) is a highly damaging ROS created in PSII [34,70,71,72,73], and high concentrations of ^1^O_2_ activate programmed cell death [41,42]. However, the increased photoprotective energy dissipation (Φ*_NPQ_*) after feeding at the whole leaflet level resulted in no difference in Φ*_NO_* compared to that before feeding (Figure 2a). Thus, at the whole leaflet level, there was no difference in ^1^O_2_ formation before and after feeding.

An increased NPQ formation decreases ^1^O_2_ production [74,75,76], and zeaxanthin, a pigment involved in a form of NPQ, may also directly quench ^1^O_2_ [77]. Thus, the increased NPQ at the whole leaflet level, after 15 min feeding (Figure 3a), might have contributed to the observed similarity in ^1^O_2_ formation before and after feeding. The photoprotective dissipation of excess light energy as heat (NPQ) can be considered to be efficient, under biotic or abiotic stress conditions, only if it is regulated in such a way so as to maintain the same fraction of open reaction centers as that in control conditions [49,78]. In our experiment, the NPQ increase at the whole leaflet after feeding (Figure 3a) was not adequate, since the fraction of open reaction centers at the whole leaflet did not remain open to the same level as before feeding (Figure 2b).

Several studies have demonstrated inhibition of photosynthesis following insect herbivory [19,79,80,81] and within this context, ROS play an essential role [15]. A quick transient production of ROS, characterized as an “oxidative burst”, is a mark of successful recognition of plant herbivory [82,83]. The harmful oxidations at the feeding zone were probably the result of ^1^O_2_ formation via the ^3^chl* (Figure 2a) and H_2_O_2_ generation (Figure 6). ROS are formed simultaneously by energy transfer (^1^O_2_) and electron transport (H_2_O_2_) [84]. ROS production at high levels was originally defined as lethal for the cell, but later on high ROS production was recognized also as necessary for plant defense (oxidative burst, necrosis). More recently, it has been demonstrated that ROS are involved as signal molecules during cellular growth to control stomatal closure, in programmed cell death. and in biotic and abiotic stress responses in plants [25,85,86,87,88].

Plants have developed mechanisms to control the creation and scavenging of ROS through antioxidative processes [28,43,89,90,91]. ROS have a double function in biotic and abiotic stresses, being beneficial for triggering defense responses by activating local and systemic plant defense responses at low levels, while at elevated levels out of the boundaries, they are harmful to cells [41,43,91]. Hydrogen peroxide generation was noticed only at the feeding zone and did not spread out to the rest of the leaf (Figure 6). It has been frequently observed to diffuse through the leaf veins acting as a molecule that triggers a long-distance stress defense response [24,28,36,38,41,52,84] or induces programmed cell death in plants [41,42]. In contrast to the local production of ROS, the increased NPQ at the whole leaf level may be suggested as a major component of the systemic acquired resistance [92].

According to the PSII model [68], the decreased Φ*_PSII_* at the whole leaflet after feeding compared to that before feeding (Figure 2a) can be attributed either to a decreased fraction of open PSII reaction centers (q*_p_*) or to a lower efficiency of these centers (F*v*′/F*m*′). The decreased Φ*_PSII_* after feeding at the whole leaflet level was due to a decreased fraction of open reaction centers, since the efficiency of open PSII centers to utilize the absorbed light did not differ before and after feeding at the whole leaflet level (Figure 5b). In accordance, to the decreased capacity to keep quinone (QA) oxidized [84,93,94], potato leaflets showed a high PSII excitation pressure (Figure 5a). High excitation pressure expresses excess energy and consequently an imbalance between energy resources and requirements [23,84,95], resulting in growth reduction and reduced plant fitness [12].

Although our results show that insect herbivory leads to a reduction of photosynthetic function, as has been reported in the literature in most cases [10,13,14,15,16,17,18,19,96], herbivore feeding may often systemically induce an increase of photosynthesis [79,97]. Herbivore insects that eliminate the leaf tissue modify the amount of source tissue without disturbing the amount of the sink tissue, e.g., roots and stems [79]. Thus, photosynthesis of the leftover undamaged tissue at the neighboring leaf area may increase to balance for the demands of the sink tissue [79].

Although the whole leaf PSII efficiency decreased compared to that before feeding, F*v*/F*m* and F*v*/F*o* were not altered by feeding, thus they cannot be considered as sensitive parameters to monitor biotic stress effects. We can conclude that chlorophyll fluorescence imaging analysis can be used efficiently to monitor even short-term effects of insect herbivory on the photosynthetic function. Chlorophyll fluorescence imaging analysis also proved to be a good indicator for quantifying the spatiotemporal heterogeneity in the allocation of the absorbed light energy to the various paths and to reveal short-term biotic stress impacts on the mechanisms of PSII functionality. Among the studied parameters, the reduction status of the plastoquinone pool displayed the highest spatiotemporal heterogeneity, being the most sensitive and suitable indicator to probe photosynthetic function and determine the impact of biotic and abiotic stresses on plants [36,94].

## 4. Materials and Methods

### 4.1. Plant Material and Growth Conditions

Potato plants (*Solanum tuberosum* L. cv Spunta) were grown in plastic pots that contained peat moss (Terrahum) and perlite (Geoflor) (1:1 *v*/*v*) in an insect proof greenhouse, under 23 ± 3 °C day temperature, 17 ± 3 °C night temperature, 70 ± 5% relative humidity, and natural light.

### 4.2. Tuta Absoluta

Individuals used in this experiment originated from a colony of *Tuta absoluta* (Meyrick) (Lepidoptera: Gelechiidae) maintained in the Entomology Lab of the Institute of Plant Breeding and Genomic Resources (Thermi, Greece). *Tuta absoluta* were reared in a climate-controlled room at 26  °C, 60% RH, and a photoperiod of 16:8 (L:D) h. Approximately 100 adults were transferred with a simple mouth-operated aspirator (BioQuip Products, Compton, CA, USA) in pop-up breeding cages (40 × 40 × 60 cm) with a vinyl window and zip closure (Watkins & Doncaster, Leominster, UK) containing two insect-free potato plants. Adults were provided water and 10% sucrose solution and allowed to oviposit for 24–48 h. After that, infested leaves (visual observation) were carefully removed and placed inside smaller breeding cages (30 × 30 × 30 cm) to allow larval development to second instar. To ensure availability of food, infested leaves were placed on a potted potato plant. Second-instar larvae (L2) used in all bioassay experiments were starved for 12 h prior to feeding.

### 4.3. Experimental Design

In each experimental plant, the terminal leaflet of the 4th leaf was used for the measurements. The leaflet was enclosed in the measurement chamber of a fluorometer and the photosynthetic efficiency was measured. One second-instar larva (L2) was added and allowed to feed for 15 min without removing the leaflet from the fluorometer’s measurement chamber. After removing the larva, new measurements were conducted on the same leaflet immediately after feeding. Five different plants were examined, and from each plant the terminal leaflet was selected to be analyzed.

### 4.4. Chlorophyll Fluorescence Imaging Analysis

Chlorophyll fluorescence imaging analysis was conducted using an Imaging Pam Fluorometer M-Series, Mini Version (Walz, Effeltrich, Germany), as described previously [98]. Dark-adapted leaves (15 min) from five different plants were measured with the actinic light (AL) intensity of 636 μmol photons m^−2^ s^−1^. In each leaflet, 16–18 areas of interest (AOIs) were selected before herbivory, and after herbivory an AOI was added covering each spot of herbivory (feeding spot). The measured chlorophyll fluorescence parameters are shown in Table 1. The minimum chlorophyll *a* fluorescence in the light-adapted leaf (F*o*′) was computed by the Imaging Win software using the approximation of Oxborough and Baker [99] as F*o*′ = F*o*/(F*v*/F*m* + F*o*/F*m*′). Representative color code images that are displayed in Figure 1 were obtained with 636 μmol photons m^−2^ s^−1^ AL.

### 4.5. Hydrogen Peroxide Imaging

Detection of H_2_O_2_ after 15 min feeding was performed as described earlier [28]. Briefly, potato leaves were incubated with 25 μM 2′, 7′-dichlorofluorescein diacetate (DCF-DA, Sigma Aldrich, St. Louis, MO, USA) for 30 min in the dark, and observed under a Zeiss AxioImager Z2 epi-fluorescence microscope equipped with an AxioCam MRc5 digital camera [28].

### 4.6. Statistics

Statistically significant differences were determined from four independent measurements using two-way ANOVA. Means (±SD) were considered statistically different at a level of *p* < 0.05.

## Figures and Tables

**Figure 1 molecules-26-02984-f001:**
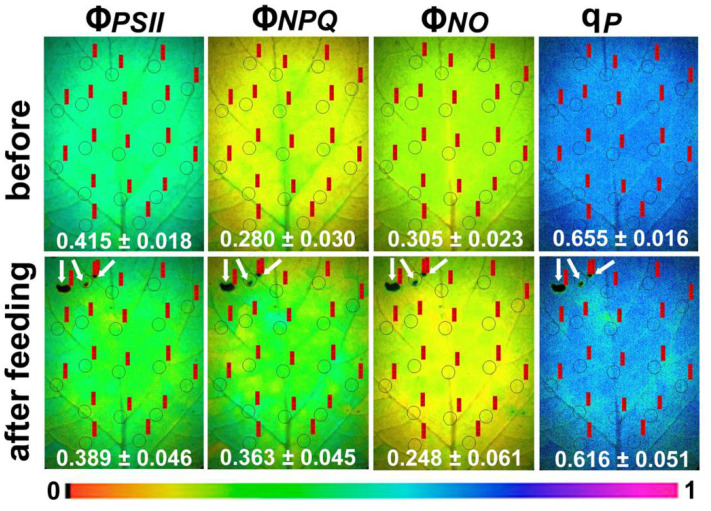
Representative images of the effective quantum yield of PSII photochemistry (Φ***_PSΙΙ_***), the quantum yield of regulated non-photochemical energy loss in PSII (Φ***_NPQ_***), the quantum yield of non-regulated energy loss in PSII (Φ***_NO_***), and the fraction of open PSII reaction centers (qp) of potato leaves before and after *Tuta absoluta* feeding. The areas of interest (AOIs) are shown in each image, with their values in red labels, together with the whole leaflet average value (±SD). The white arrows in the images show the feeding zones. The color code depicted at the bottom ranges from values 0 to 1.

**Figure 2 molecules-26-02984-f002:**
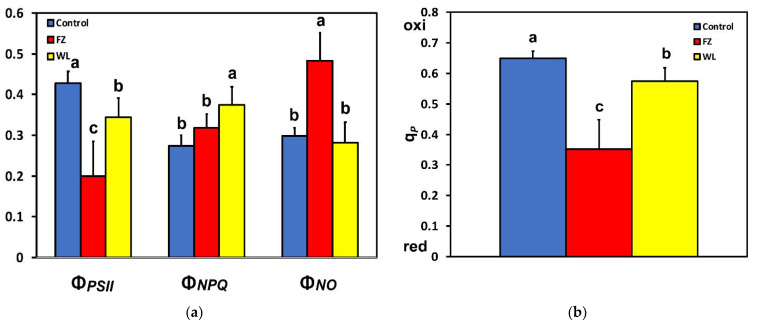
Changes in (**a**) the effective quantum yield of PSII photochemistry(Φ*_PSΙΙ_*), the quantum yield of regulated non-photochemical energy loss in PSII (Φ*_NPQ_*), and the quantum yield of non-regulated energy loss in PSII (Φ*_NO_*); and in (**b**) the fraction of open PSII reaction centers (q*_p_*) at the whole leaflet level before feeding (control), at the feeding zone after feeding (FZ), and at the whole leaflet after feeding (WL). Error bars are standard deviations. Columns with different letters are statistically different (*p* < 0.05).

**Figure 3 molecules-26-02984-f003:**
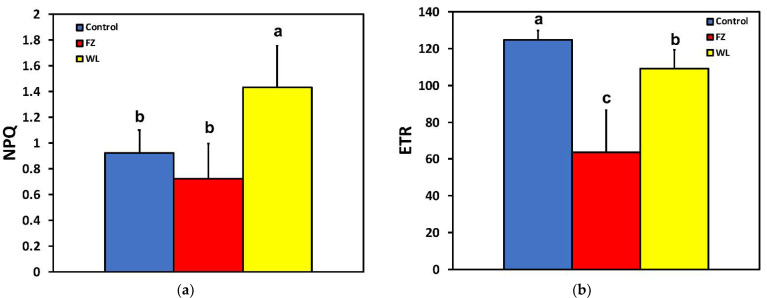
Changes in (**a**) the non-photochemical quenching (NPQ) and (**b**) in the electron transport rate (ETR) at the whole leaflet level before feeding (control), at the feeding zone after feeding (FZ), and at the whole leaflet after feeding (WL). Error bars are standard deviations. Columns with different letters are statistically different (*p* < 0.05).

**Figure 4 molecules-26-02984-f004:**
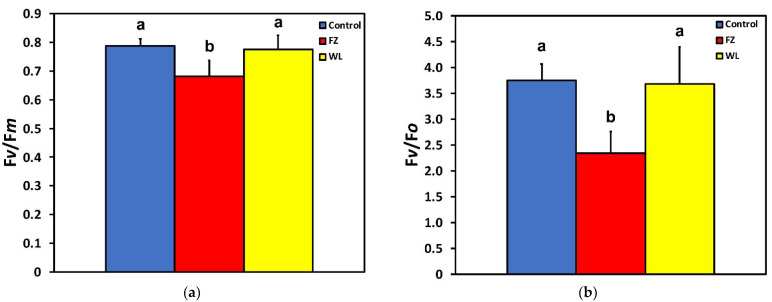
Changes in (**a**) the maximum efficiency of PSII photochemistry (F*v*/F*m*) and (**b**) in efficiency of the water-splitting complex on the donor side of PSII (F*v*/F*o*) at the whole leaflet level before feeding (control), at the feeding zone after feeding (FZ), and at the whole leaflet after feeding (WL). Error bars are standard deviations. Columns with different letters are statistically different (*p* < 0.05).

**Figure 5 molecules-26-02984-f005:**
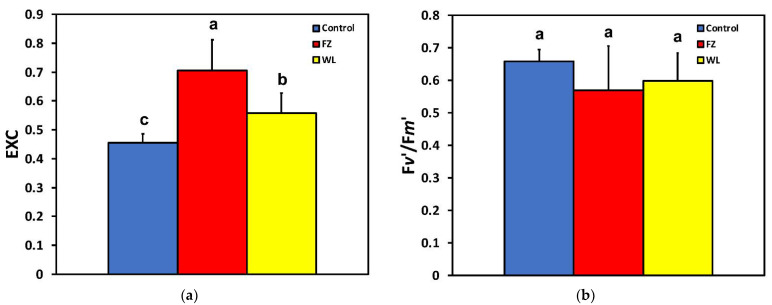
Changes in (**a**) the excess excitation energy (EXC) and (**b**) in the efficiency of open PSII reaction centers (F*v*′/F*m*′) at the whole leaflet level before feeding (control), at the feeding zone after feeding (FZ), and at the whole leaflet after feeding (WL). Error bars are standard deviations. Columns with different letters are statistically different (*p* < 0.05).

**Figure 6 molecules-26-02984-f006:**
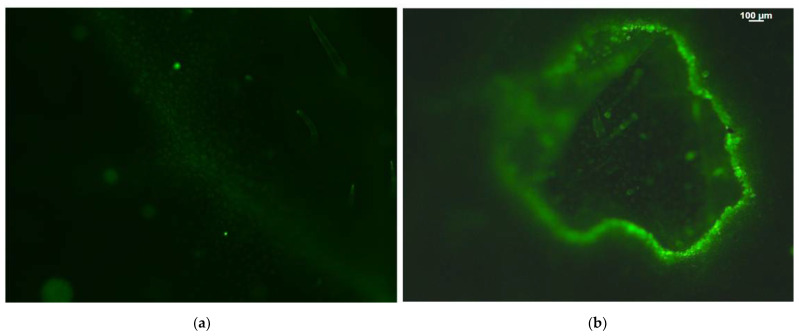
Hydrogen peroxide (H_2_O_2_) detection by staining of potato leaflets with 2′, 7′-dichlorofluorescein diacetate (DCF-DA) before (**a**) and after (**b**) *T. absoluta* feeding. The whole area of a feeding zone is shown. Increased generation of hydrogen peroxide is visible by the light green color. Scale Bar: 100 μm.

**Table 1 molecules-26-02984-t001:** Definitions of the measured chlorophyll fluorescence parameters.

Parameter	Definition	Calculation
F*v*/F*m*	Maximum efficiency of PSII photochemistry	(F*m* − F*o*)/F*m*
F*v*/F*o*	Efficiency of the water-splitting complex on the donor side of PSII	(F*m* − F*o*)*/*F*o*
Φ*_PSII_*	Effective quantum yield of PSII photochemistry	(F*m*′ − F*s*)/F*m*′
Φ*_NPQ_*	Quantum yield of regulated non photochemical energy loss in PSII	F*s*/F*m*′ − F*s*/F*m*
Φ*_NO_*	Quantum yield of non-regulated energy loss in PSII	F*s*/F*m*
q*_p_*	Photochemical quenching, representing the redox state of the plastoquinone pool or the fraction of open PSII reaction centers	(F*m*′ − F*s*)/(F*m*′ − F*o*′)
ETR	Electron transport rate	ΦPSII × PAR × c × abs, where PAR is the photosynthetically active radiation, c is 0.5, and abs is the total light absorption of the leaf taken as 0.84
NPQ	Non-photochemical quenching reflecting the dissipation of excitation energy as heat	(F*m* − F*m*′)/F*m*′
EXC	Excess excitation energy	(F*v*/F*m* − Φ*_PSII_*)/(F*v*/F*m*)
Fv′/Fm′	Efficiency of open PSII reaction centers	(F*m*′ − F*o*′)/F*m*′

## Data Availability

Not applicable.

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
