# Peer review of "Changes in Light Energy Utilization in Photosystem II and Reactive Oxygen Species Generation in Potato Leaves by the Pinworm Tuta absoluta"

_molecules, 2021, doi:10.3390/molecules26102984_

Round 1
Reviewer 1 Report
Nice work. In my opinion the reported findings are important by using chlorophyll fluorescence image analysis. I think some minor changes might be done in the text and some figures before final acceptance of this submission (see attached file).

Author Response
All of the comments were answered on the Reviewer’s pdf file.

Reviewer 2 Report
I noticed large number of references were used in citation. My advice to be reduced to keep only the most important and significant once
Author Response
I noticed large number of references were used in citation. My advice to be reduced to keep only the most important and significant once.
We rather agree with your comment about the number of references used, but it will be a dilemma now to decide which one to delete and which one to keep. Yet, deleting some references sometimes results in mistakes in the numbers of the remaining citations in the reference list. We will try in our next manuscript to have only the most important and significant citations.
Reviewer 3 Report
The authors have evaluated the chlorophyll fluorescence of photosystem II in detached potato leaflets before and after feeding by Tuta absoluta leaf miner. They observed that the plastoquinone pool of the leaflet was more reduced after feeding by T. absoluta. They also showed that singlet oxygen and hydrogen peroxide formation increased at the feeding site while non-photochemical quenching increased at the whole leaflet level.
There is some inconsistency between text and figures:
Line 111, the authors write that there is a significant increase of ΦNO (Figure 2a), which is clearly opposite to what can be seen in Figure 1. And in the next line (113), the authors indicate that there was no significant change of ΦNO !! By the way, the values in red label (line 118) are absent.
The authors could also suggest mechanisms of action that led to ROS production locally and to systemic NPQ in the leaflet.
In the section Materials and Methods, larvae were allowed to feed for 20 minutes (line 273) but for 15 minutes only in the rest of the article.
Author Response
The authors have evaluated the chlorophyll fluorescence of photosystem II in detached potato leaflets before and after feeding by Tuta absolutaleaf miner. They observed that the plastoquinone pool of the leaflet was more reduced after feeding by T. absoluta. They also showed that singlet oxygen and hydrogen peroxide formation increased at the feeding site while non-photochemical quenching increased at the whole leaflet level.
There is some inconsistency between text and figures:
Line 111, the authors write that there is a significant increase of ΦNO(Figure 2a), which is clearly opposite to what can be seen in Figure 1. And in the next line (113), the authors indicate that there was no significant change of ΦNO !! By the way, the values in red label (line 118) are absent.
We have re-written the above lines so that to avoid any confusion that can allow the reader to think of an inconsistency. Thank you for pointing the problem in these lines so that to clarify it. In line 111 the significant increase of ΦNOrefers to the feeding site (Figure 2a) while in Figure 1 the value refers to the whole leaf. Despite the lower ΦNO value shown in Figure 1 for the whole leaf there was no significant difference between control (before feeding) and after feeding from mean values of the 5 leaves analysed at the whole leaf level.
The values in red labels (line 118) are not absent but they are not easy distinguishable in the word format or in the pdf format. In the original figures that we uploaded together with the manuscript in tif format can been seen after magnification. They can also be seen only in the html format of the published article after magnification of the Figure. For example, you can see the relevant Figures in our previous articles in pdf format and in html format, (https://www.mdpi.com/2223-7747/9/8/962/htmand https://www.mdpi.com/2223-7747/10/3/521/htm) where you can read the values only in the html format after magnification.
The authors could also suggest mechanisms of action that led to ROS production locally and to systemic NPQ in the leaflet.
In lines 206-207 we suggest that “The harmful oxidations at the feeding zone, were probably the result of 1O2formation via the 3chl* (Figure 2a) and H2O2generation (Figure 6).”, and in lines 218-219 “Hydrogen peroxide generationwas noticed only at the feeding zone, not spreading out to the rest of the leaf (Figure 6).”, suggesting a local ROS production. We also added in lines 222-223 “In contrast to the local production of ROS, the increased NPQ at the whole leaf level may be suggestedas a major component of the systemic acquired resistance [92].
In the section Materials and Methods, larvae were allowed to feed for 20 minutes (line 273) but for 15 minutes only in the rest of the article.
Thank you for pointing out our mistake. We corrected it to 15 min (line 275).